# The Effects of CeO₂ and Co Doping on the Properties and the Performance of the Ni/Al₂O₃-MgO Catalyst for the Combined Steam and CO₂ Reforming of Methane Using Ultra-Low Steam to Carbon Ratio

**Nichthima Dharmasaroja [1,2]** , **Tanakorn Ratana [1,2], Sabaithip Tungkamani [1,2],**
**Thana Sornchamni [3] and David S. A. Simakov [4] and Monrudee Phongaksorn [1,2],***

[1]  Department of Industrial Chemistry, King Mongkut's University of Technology North Bangkok, Bangkok 10800, Thailand; nichthima.d@gmail.com (N.D.); tanakorn.r@sci.kmutnb.ac.th (T.R.); sabaithip.t@sci.kmutnb.ac.th (S.T.)

[2]  Research and Development Center for Chemical Engineering Unit Operation and Catalyst Design (RCC), King Mongkut's University of Technology North Bangkok, Bangkok 10800, Thailand

[3]  PTT Research and Technology Institute, PTT Public Company Limited, Wangnoi, Ayutthaya 13170, Thailand; thana.s@pttplc.com

[4]  Department of Chemical Engineering, University of Waterloo, Waterloo, ON N2L 3G1, Canada; dsimakov@uwaterloo.ca

*  Correspondence: monrudee.p@sci.kmutnb.ac.th; Tel.: +66-255-52000 (ext. 4822); Fax: +66-258-78251

**Abstract:** In this paper, the 10 wt% Ni/Al₂O₃-MgO (10Ni/MA), 5 wt% Ni-5 wt% Ce/Al₂O₃-MgO (5Ni5Ce/MA), and 5 wt% Ni-5 wt% Co/Al₂O₃-MgO (5Ni5Co/MA) catalysts were prepared by an impregnation method. The effects of CeO₂ and Co doping on the physicochemical properties of the Ni/Al₂O₃-MgO catalyst were comprehensively studied by N₂ adsorption-desorption, X-ray diffraction (XRD), transmission electron microscopy (TEM), H₂ temperature programmed reduction (H₂-TPR), CO₂ temperature programmed reduction (CO₂-TPD), and thermogravimetric analysis (TGA). The effects on catalytic performance for the combined steam and CO₂ reforming of methane with the low steam-to-carbon ratio (S/C ratio) were evaluated at 620 °C under atmospheric pressure. The appearance of CeO₂ and Co enhanced the oxygen species at the surface that decreased the coke deposits from 17% for the Ni/MA catalyst to 11–12% for the 5Ni5Ce/MA and 5Ni5Co/MA catalysts. The oxygen vacancies in the 5Ni5Ce/MA catalyst promoted water activation and dissociation, producing surface oxygen with a relatively high H₂/CO ratio (1.6). With the relatively low H₂/CO ratio (1.3), the oxygen species at the surface was enhanced by CO₂ activation-dissociation via the redox potential in the 5Ni5Co/MA catalyst. The improvement of H₂O and CO₂ dissociative adsorption allowed the 5Ni5Ce/MA and 5Ni5Co/MA catalysts to resist the carbon formation, requiring only a low amount of steam to be added.

**Keywords:** combined steam and CO₂ reforming of methane; bimetallic catalysts; metal dispersion; coking resistance; syngas production; hydrogen production

---

## 1. Introduction

The combined steam and CO₂ reforming of methane (CSCRM) (Equation (1)) is a process that combines steam reforming of methane (Equation (2)) and CO₂ reforming of methane (Equation (3)) in one process. The CSCRM has received noticeable attention as it consumes two main greenhouse gases (CH₄ and CO₂) with water vapor to produce the synthesis gas (a mixture of H₂ and

CO) [1–4]. Although CSCRM can control an $H_2/CO$ ratio of 2 in the syngas product by the inlet feed composition [2,5,6], the obstacles for a commercial CSCRM consist of the catalyst deactivation and the energy consumption. Main causes of the catalyst deactivation are the coke deposition, which is a by-product from side reactions (the $CH_4$ decomposition (Equation (4)) and the Boudouard reaction (Equation (5)) [7–9]. To prevent the formation of carbon, steam in the feed must be sufficient. However, the quality of energy consumption can be over expected due to the evaporation of water. Therefore, the development of high-performance catalysts for CSCRM operating at a low steam-to-carbon ratio could be a critical challenge of syngas production technologies.

The non-noble metal catalysts, Ni-based catalysts, have focused the high catalytic performance and low cost as compared to the noble metal [1,10–14]. According to numerous works on the catalyst development, Ni-bimetallic catalysts have been proposed as carbon tolerance catalysts for methane reforming. Noble metals and non-noble metals are commonly used with Ni for this type of catalysts [15,16]. Among several metals, the oxygen vacancy sites can be created when $Ce^{4+}$ in the oxide form of cerium ($CeO_2$) transforms to $Ce^{3+}$. The released oxygen then removes the carbonaceous species on the Ni surface, suppressing the coke formation [7,17–20]. Furthermore, the strong interaction between Ni and Ce prevents the agglomeration of Ni nanoparticles in the Ni-Ce/montmorillonite catalyst [21]. Shan et al. [22] suggested that $Ni^{2+}$ ions can be incorporated into the lattice of $CeO_2$ and oxygen vacancies that are simultaneously generated. The presence of $CeO_2$ decreases the formation of $NiAl_2O_4$ due to the formation of $CeAlO_3$ and $Ce_{1-X}Ni_XO_2$ [23–25]. Moreover, $CO_2$ can transform into carboxylate species and react with surface hydroxyls to produce formate ($HCOO^-$) species on $Ce^{3+}$ that promoted the water gas shift reaction [25].

Cobalt (Co) is also an alternative metal component for bimetallic Ni catalysts because of the cobalt oxide properties. Co as an oxide form possess a weak metal-oxygen bond strength with high turnover frequency for a redox reaction via the reduction pathway of $Co_3O_4 \rightarrow CoO \rightarrow Co°$ [26]. Thus, cobalt oxides ($CoO_x$) play a major role in the oxidation of carbon species via redox reactions that decrease the carbon deposition [27]. During reduction, the combination of Ni-Co metal creates the formation of the Ni-Co alloy structure [11]. The Ni-Co alloy enhances the prevention of crystal growth, the adsorption of oxygen species, and the distribution of active sites, resulting in higher reactant conversions [28]. Moreover, the partial Co metal diffusing from the bulk structure to the spinel-like phases ($CoAl_2O_4$) deducts the sintering of active metal during the reaction [29].

The reactant ratio adjustment in the feed plays a vital role in the carbon deposition control. Although $CH_4$ is the source of hydrogen for $H_2$ production, $CH_4$ dissociation is the main reaction of coke formation, especially under the relatively low $CO_2$ content conditions. Certain research articles concluded that the satisfaction ratio of $CO_2$ to $CH_4$ ($CO_2/CH_4$) is greater than unity [26]. Steam in the feed promotes the steam reforming of methane as well as the water gas shift reaction, which increases the $H_2/CO$ ratio in the syngas product. The addition of $H_2O$ molecules also provides more of an oxygen source for the carbon removal mechanism (Equation (6)) [11,27,28,30]. The extra energy to evaporate a large volume of steam at the inlet and to separate water from the gaseous outlet is concerning [11,29].

$$3CH_4 + CO_2 + 2H_2O \rightarrow 8H_2 + 4CO \qquad \triangle H°_{298\,K} = +659 \text{ kJ/mole} \qquad (1)$$

$$CH_4 + H_2O \rightarrow 3H_2 + CO \qquad \triangle H°_{298\,K} = +206 \text{ kJ/mole} \qquad (2)$$

$$CH_4 + CO_2 \rightarrow 3H_2 + CO \qquad \triangle H°_{298\,K} = +247 \text{ kJ/mole} \qquad (3)$$

$$CH_4 \rightarrow C + 2H_2 \qquad \triangle H°_{298\,K} = +74.8 \text{ kJ/mole} \qquad (4)$$

$$2CO \rightarrow C + CO_2 \qquad \triangle H°_{298\,K} = -173.3 \text{ kJ/mole} \qquad (5)$$

$$C + H_2O \rightarrow H_2 + CO \qquad \triangle H°_{298\,K} = +131.3 \text{ kJ/mole} \qquad (6)$$

This work evaluated the effect of $CeO_2$ and Co promoters over the CSCRM (using a low steam- to-carbon ratio (S/C ratio) that accompanies $CO_2$ and $H_2O$ oxidants) catalytic performance

of the Ni/MgO-Al$_2$O$_3$. For this propose, 10 wt% Ni/MgO-Al$_2$O$_3$ (10Ni/MA), 5 wt% Ni–5 wt% Ce/MgO-Al$_2$O$_3$ (5Ni5Ce/MA), and 5 wt% Ni–5 wt% Co/MgO-Al$_2$O$_3$ (5Ni5Co/MA) were prepared by impregnation methods. The CSCRM performances of all catalyst samples were investigated at 620 °C under atmospheric pressure. The carbon accumulation on the surface of the spent catalysts was investigated using thermogravimetric analysis (TGA). The correlation between catalytic performance and properties, characterized by N$_2$ adsorption-desorption, X-ray diffraction (XRD), transmission electron microscope (TEM), H$_2$ temperature programmed reduction (H$_2$-TPR), and CO$_2$ temperature programmed desorption (CO$_2$-TPD), were revealed.

## 2. Results and Discussion

The crystalline phases of the calcined catalysts were analyzed by XRD (Figure 1). In all catalysts, the diffraction peaks of meixnerite Mg$_6$Al$_2$(OH)$_{18}$·4H$_2$O were indicated at 2θ of 11.5°, 22.5°, and 35.0° [31]. The characteristic diffraction peaks at 2 theta of 36.8°, 44.8°, and 65.3° can be assigned to the spinel phase of support (MgAl$_2$O$_4$), existing as an overlap with the NiAl$_2$O$_4$ (or CoAl$_2$O$_4$) spinel. The presence of NiAl$_2$O$_4$ and CoAl$_2$O$_4$ spinel reflects the strong metal-support interaction in the catalysts. The board peaks of CeO$_2$ diffractogram at 2θ of 28.5°, 47.5°, and 56.3° were observed on the 5Ni5Ce/MA catalyst. In the 5Ni5Co/MA catalyst, the peaks of the Co$_3$O$_4$ phase at 2θ of 19.3° and 31.5° were detected. The discrete peaks located at 2θ = 37.0°, 43.0°, 62.4°, 74.8°, and 78.7° corresponding to the NiO phase overlapped with MgO peaks [32].

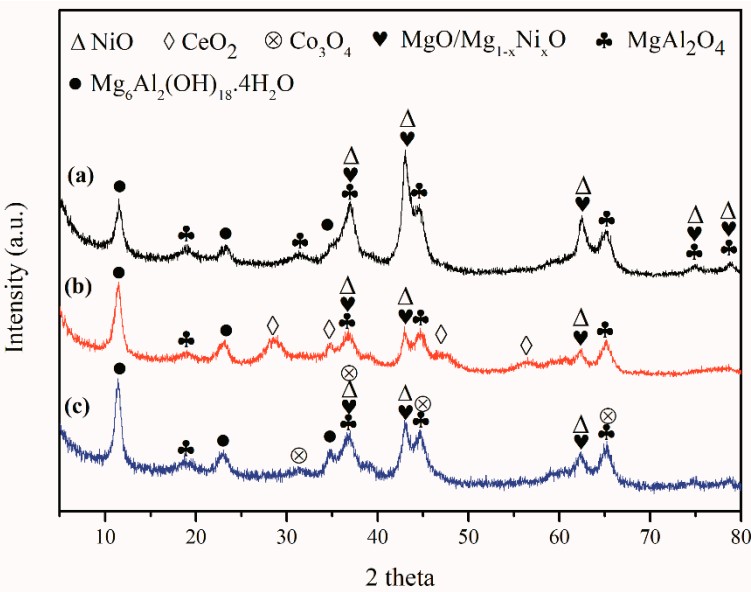

**Figure 1.** Powder XRD patterns of fresh 10Ni/MA (**a**), 5Ni5Ce/MA (**b**), and 5Ni5Co/MA (**c**) catalysts.

The N$_2$ adsorption-desorption isotherms and the pore size distribution of 10Ni/MA, 5Ni5Ce/MA, and 5Ni5Co/MA catalysts are shown in Figure 2. The isotherms of all catalysts are categorized as type IV according to the International Union of Pure and Applied Chemistry (IUPAC) classification and the pore sizes of samples are mainly in the range of 4.6–6.1 nm, implying the characteristic of mesoporous materials. Hysteresis loops of types H1 and H3 corresponding to the combination of cylindrical pores and parallel plate-shaped pores are found in all catalysts [33]. The surface area and total pore volume of samples are listed in Table 1. Compared to the monometallic Ni supported catalyst (10Ni/MA), bimetallic catalysts (5Ni5Ce/MA and 5Ni5Co/MA) provided a larger catalyst surface and pore volume. TEM pictures of all samples presented in Figure 3 suggests a better dispersion of the metal species with a smaller average metal size on 5Ni5Ce/MA and 5Ni5Co/MA catalysts as compared to the 10Ni/MA catalyst, which agree with N$_2$ adsorption-desorption results. It can be explained that the presence of

CeO$_2$ and Co$_3$O$_4$ in Ni/MA reduces the NiO agglomeration, resulting in the increase in Ni dispersion, surface area, and the pore volume.

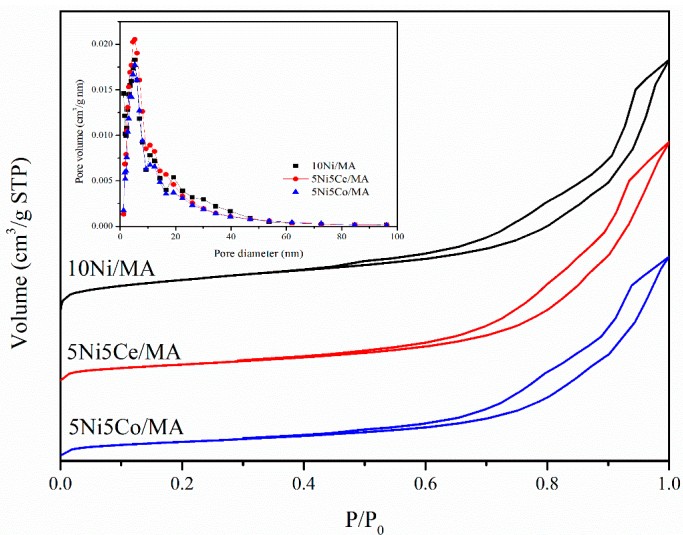

**Figure 2.** Nitrogen adsorption–desorption isotherms including the corresponding pore size distribution curves (inset) of 10Ni/MA, 5Ni5Ce/MA, and 5Ni5Co/MA catalysts.

**Table 1.** The properties of 10Ni/MA, 5Ni5Ce/MA, and 5Ni5Co/MA catalysts.

| Catalyst | N$_2$ Adsorption-Desorption [a] | | H$_2$-TPR [b] | Deconvolution of the CO$_2$-TPD [b] (mmol/g$_{cat}$) | | | |
| | Surface Area (m$^2$/g) | Total Pore Volume (cm$^3$/g) | H$_2$ Uptake (mmol/g$_{cat}$) | Weak | Medium | Strong | Total Basicity |
|---|---|---|---|---|---|---|---|
| 10Ni/MA | 97 | 0.20 | 11.9 | 0.04 | 0.02 | 0.13 | 0.19 |
| 5Ni5Ce/MA | 111 | 0.24 | 4.3 | 0.02 | 0.03 | 0.17 | 0.22 |
| 5Ni5Co/MA | 108 | 0.22 | 10.8 | 0.03 | 0.03 | 0.10 | 0.16 |

[a] The systematic error of N$_2$ adsorption-desorption is ±5%. [b] The systematic error is ±1% for temperature and ±8% for the quantity of gaseous substances.

The TPR profile of the 10Ni/MA catalyst (Figure 4) showed three peaks associated with the reduction of NiO located on the surface of MA support (a sharp reduction peak at 350 °C), the reduction of NiO nanoparticles confined in the MA support (a peak shoulder at 450 °C), and the reduction of Mg(Ni,Al)O (the main peak at 790 °C), which is a form of the strong metal-support interaction [34,35]. The 5Ni5Ce/MA catalyst revealed four reduction board peaks at 295 °C, 410 °C, 530 °C, and 845 °C. The lowest temperature peak is preliminarily ascribed to the oxygen mobility on the surface catalyst that substitutes the Ni ion incorporated with CeO$_2$ at the surface and the reduction of Ce$^{4+}$ located on the surface [24,36,37]. The second and third peaks can be tentatively assigned to the reduction of Ni-CeO$_x$ solid solution. The highest temperature peak could be the reduction peak of Ce species simultaneously with the reduction peak of Ni species strongly interacting with the support. Compared to the TPR profile of the 10N/MA, the reduction peak at high temperature in the TPR profile of 5Ni5Ce/MA shifted to higher temperature because the smaller Ni particle size in the 5Ni5Ce/MA catalyst led to the stronger metal-support interaction [38]. As seen in the TPR profile of 5Ni5Co/MA, the low-temperature signal at about 270 °C corresponded to the simultaneous reduction of NiO to Ni and Co$_3$O$_4$ to CoO. The shoulder at 370 °C related to the reduction of CoO to Co species [39]. The peak at 790 °C starting at about 500 °C is attributed to the reduction of NiAl$_2$O$_4$ or the CoAl$_2$O$_4$ spinel phase. The reduction peak at low temperature in the TPR profile of 5Ni5Co/MA catalyst shifted to a lower temperature compared to 10Ni/MA. It implied to the weak interaction between metal and support, which can be attributed to the formation of the Ni–Co alloy [40–43].

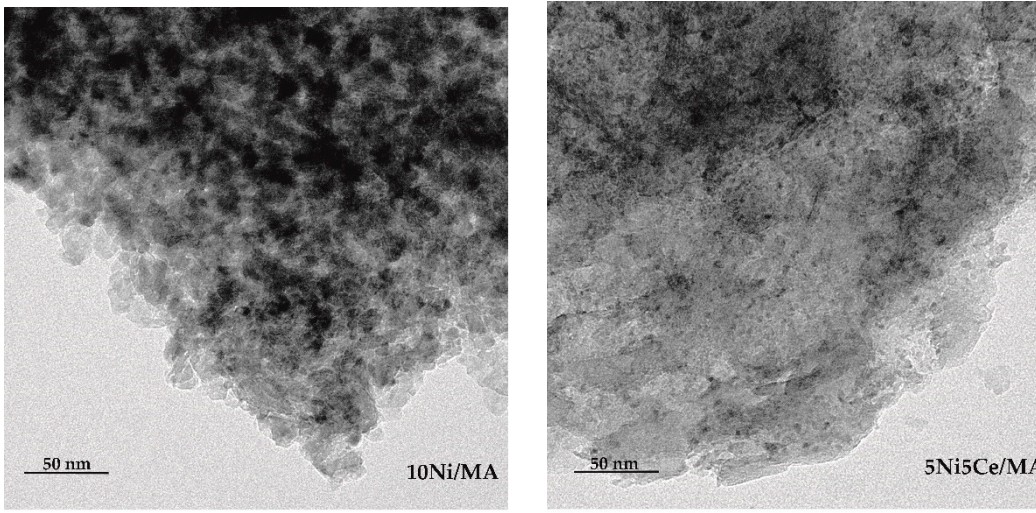

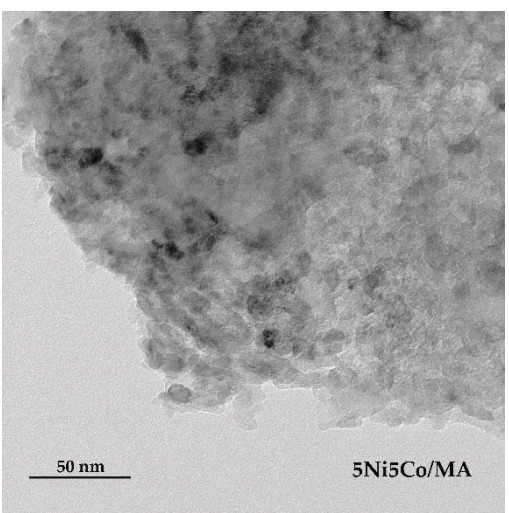

**Figure 3.** TEM images of 10Ni/MA, 5Ni5Ce/MA, and 5Ni5Co/MA catalysts.

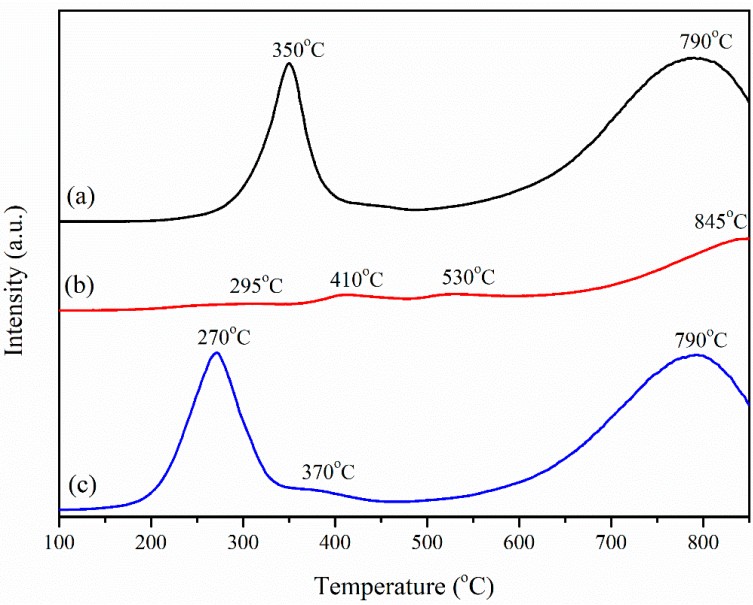

**Figure 4.** H$_2$-TPR profile of 10Ni/MA (a), 5Ni5Ce/MA (b), and 5Ni5Co/MA (c) catalysts.

The strength and quantity of basic sites on the surface of catalysts were evaluated from the $CO_2$ desorption temperature and the desorption areas of peaks in the $CO_2$-TPD profiles (Figure 5), respectively. Figure 5 reflected three types of basic sites. In each $CO_2$-TPD profile of the reduced catalysts, the peak at the lowest desorption temperature represents the weakly chemisorbed $CO_2$ on basic sites connected to the Brönsted hydroxyl groups. The peak at the middle temperature refers to medium base sites comprised to the Lewis acid-base pairing, and the highest temperature peak is assigned to strong basic sites related to the low-coordination surface oxygen ($O^{2-}$) anions [44,45]. The total number of basic sites of 10Ni/MA, 5Ni5Ce/MA, and 5Ni5Co/MA catalysts were 0.19, 0.22, and 0.16 mmol/gcat (Table 1), respectively. The 5Ni5Ce/MA catalyst showed the highest number of basic sites because of the oxygen mobility from the redox property of Ce metal. Among all catalysts, 5Ni5Ce/MA and 5Ni5Co/MA provided a greater number of medium basic sites (correlating to the oxophilicity of the surface metal in the catalyst). The oxygen mobility and the stronger oxophilicity suggest more coverage of O* species that can improve the removal of coke deposition [45].

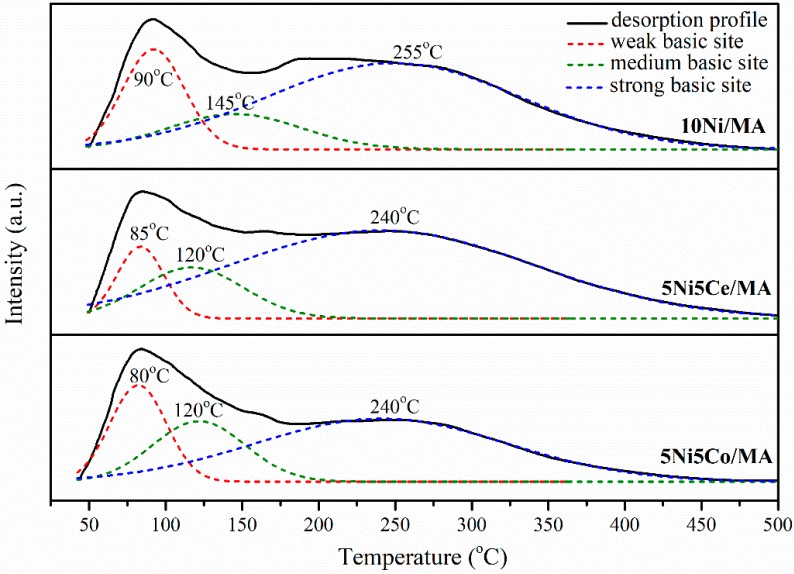

**Figure 5.** $CO_2$-TPD profile of catalysts.

CSCRM tests operated with the low steam to carbon ratio (S/C = 0.28) at isothermal condition of 620 °C under the ambient pressure for 6 h were demonstrated. The catalytic performance of the catalyst samples was evaluated in term of the $CH_4$ and $CO_2$ conversions as well as the $H_2$/CO ratio (Figure 6a–c). The results of the 10Ni/MA catalyst can be separated into two periods of time-on-steam described as part I and II. The part I (0–125 min) represented the different conversions for $CH_4$ and $CO_2$ with a low $H_2$/CO ratio (<1). Compared to part I, part II (125–360 min) showed the higher $CH_4$ conversion, the lower $CO_2$ conversion, and the higher $H_2$/CO ratio (>1). It referred that the $CO_2$ reforming of methane dominated the overall process at the initial time. An increase of $CH_4$ conversion with a decrease of $CO_2$ conversion (part II) indicated that the consumed $CH_4$ reacted with steam and $CO_2$ because the steam and $CO_2$ acted as a co-oxidant [46,47], resulting in a higher $H_2$/CO ratio of $1 < \times < 1.6$. This evidence reflected the sufficient time for $H_2O$ dissociative adsorption for the low S/C condition. Considering the performance of the 5Ni5Ce/MA catalyst, $CH_4$ conversion was less by half compared to the 10Ni/MA because of the decrease in active metal content. The 5Ni5Ce/MA catalyst showed the lowest $CO_2$ conversion with the highest $H_2$/CO ratio, suggesting the most occurrence of steam reforming of methane during the CSCRM process on 5Ni5Ce/MA among these catalysts. It can be explained that the oxygen mobility in 5Ni5Ce/MA enhances the water association-dissociation ($Ce_2O_3 + H_2O \rightarrow CeO_2 + H_2$) on the surface of the catalyst [21,48,49]. Moreover, $CO_2$ conversions at 90–120 min of the 5Ni5Ce/MA catalyst (relatively fast $H_2O$ activation) were increased when compared

to the initial time as $H_2O$ should be insufficient at a moment (for this low S/C case), resulting in the $H_2/CO$ ratio (<1). The resulting trends (conversions and $H_2/CO$ ratio) of the 5Ni5Co/MA catalyst were similar to the 10Ni/MA catalyst and reactant conversions were lower than the 10Ni/MA. For $CH_4$ conversions, the cobalt metal was less active for $CH_4$ dissociation than Ni metal because of the higher activation energy of $CH_4$ dissociation [50,51]. In part I, the difference between $CO_2$ and $CH_4$ conversions of 5Ni5Co/MA and 10Ni/MA were similar as well as the $H_2/CO$ ratios, indicating the same magnitude of $CO_2$ reforming of methane domination in the process. In part II, the $CO_2$ and $CH_4$ conversions of 5Ni5Co/MA were closer than those of 10Ni/MA and the 5Ni5Co/MA catalyst illustrated the lowest $H_2/CO$ ratios. These results expressed the highest magnitude of $CO_2$ reforming of methane domination on the 5Ni5Co/MA catalyst after 125 min time-on-stream. Li et al. [50] reported that the Co metal surface promotes more dissociative adsorption of $CO_2$ than the Ni metal surface due to the oxophilic property of Co, which is in good agreement with $CO_2$-TPD results. It also reveals that the $H_2O$ association-dissociation on cobalt metal was complicated, resulting in the lowest $H_2/CO$ ratio.

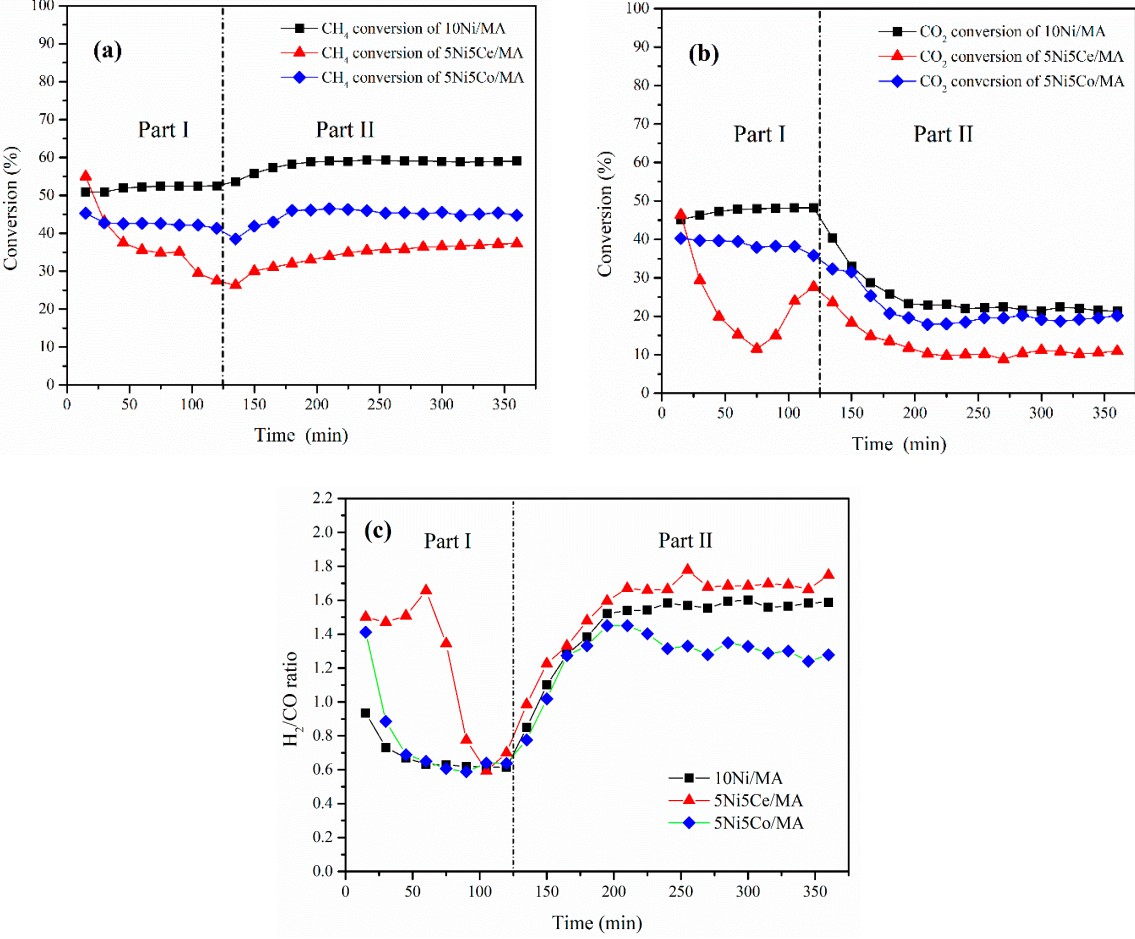

**Figure 6.** The $CH_4$ (**a**) and $CO_2$ conversions (**b**) for CSCRM and $H_2/CO$ ratios (**c**) of 10Ni/MA, 5Ni5Ce(T)/MA, and 5Ni5Ce/MA catalysts at 620 °C under ambient pressure (the relative error is about 3%).

The quantity and the types of carbon deposition on the spent catalysts were elucidated using the TGA results (Figure 7). The percentage of weight loss directly relates to the amount of carbon deposition. TGA profiles of the spent catalysts represented the physically adsorbed water and amorphous carbon (≤250 °C), the graphitic carbon (250 °C–450 °C), and the carbon filament (≥450 °C) on the surface [18,52–54]. The last two types, which are not easily oxidizable, have been considered as the major reason for the catalyst deactivation. The total weight loss of spent 10Ni/MA, 5Ni5Ce/MA, and 5Ni5Co/MA catalysts were 17%, 11%, and 12%, respectively. Although the main type of coke

in all catalysts was the graphitic carbon, the highest temperature of carbon removal was found from the spent 10Ni/MA catalyst. It implied that carbon deposition was formed more easily via the $CH_4$ dissociation/decomposition on the Ni/MA catalyst. The coke deposited can be deduced by the oxygen intermediate in $H_2O$ activation-dissociation for 5Ni5Ce/MA and oxygen intermediates in $CO_2$ activation-dissociation for 5Ni5Co/MA. Consequently, with Ce and Co promoters, the carbon tolerance of Ni/MA catalyst was improved, requiring only a small amount of steam to be added in the feed of the methane, reforming in order to prevent carbon deposition [55–57].

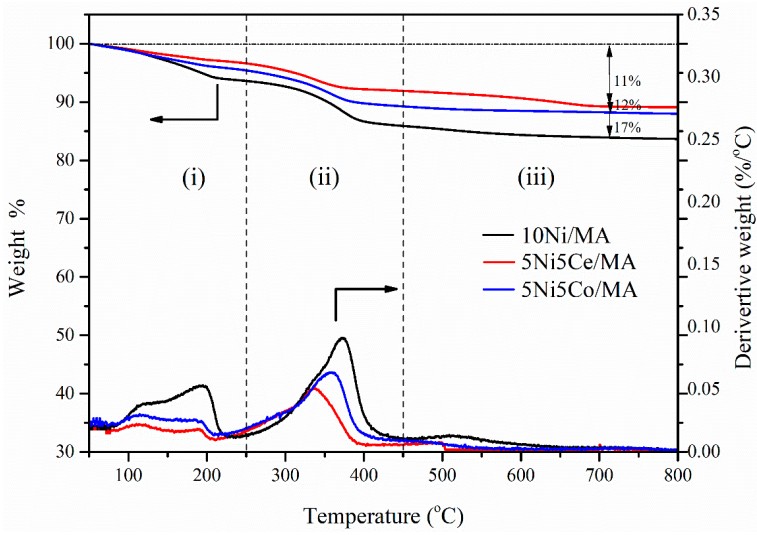

**Figure 7.** Thermogravimetric analysis (TGA) profile of spent catalysts.

A simplified reaction mechanism of CSCRM to produce synthesis gas on a Ni-based catalyst consists of three element reactions: $CH_4$ dissociation/decomposition (Equation (7)), $H_2O$ activation-dissociation (Equation (8)), and $CO_2$ activation-dissociation (Equation (9)). $CH_4$ dissociation/decomposition produces carbon surface (C*) and $H_2$. $CO_2$ and $H_2O$ activation-dissociation generate an oxygen surface (O*) with CO and $H_2$, respectively. The C* on the surface can be removed by reacting with the O* surface to form CO (Equation (10)). Therefore, the catalytic behavior of the catalyst for the reaction steps can be tentatively evaluated by conversions of reactants, $H_2/CO$ ratio, and the quantity of coke formation. The activity toward the $CH_4$ dissociation/decomposition step directly relates to $CH_4$ conversion. The $H_2O$ and $CO_2$ activation-dissociation can be preliminary predicted using the $H_2/CO$ ratio with the amount of coke. If the high $H_2/CO$ ratio with low carbon accumulation occur, the catalyst is selective for $H_2O$ activation-dissociation. Likewise, the catalyst is selective for $CO_2$ activation-dissociation when the low $H_2/CO$ ratio and low carbon accumulation are founded. According to the results under the low S/C ratio condition of CSCRM process, it suggested that 10Ni/MA is the most active catalyst for $CH_4$ dissociation/decomposition. $CeO_2$ in 5Ni5Ce/MA and Co in 5Ni5Co/MA boosts the $H_2O$ and $CO_2$ activation-dissociation, respectively.

$$CH_4 + * \rightarrow 2H_2 + C* \tag{7}$$

$$CO_2 + * \rightarrow CO + O* \tag{8}$$

$$H_2O + * \rightarrow H_2 + O* \tag{9}$$

$$C* + O* \rightarrow CO \tag{10}$$

## 3. Materials and Methods

### 3.1. Catalyst Preparation

The $Al_2O_3$-MgO (MA) was synthesized by the sol-gel method. The mixture of aluminium isopropoxide (Acros Organics$^{TM}$, Morris County, NJ, USA 98%), nitric acid (CARLO ERBA Reagents, Val-de-Reuil, France 65%), magnesium ethoxide (Sigma-Aldrich, St. Louis, MO, USA 98%), and distilled water were refluxed at 80 °C for 20 h. The resulting gel was dried at 60 °C for 24 h and calcined at 800 °C for 6 h. The MA support was crushed and sieved to 355–710 μm. The 10 wt% Ni/$Al_2O_3$-MgO (10Ni/MA), 5 wt% Ni-5 wt% Ce/$Al_2O_3$-MgO (5Ni5Ce/MA), and 5 wt% Ni-5 wt% Co/$Al_2O_3$-MgO (5Ni5Co/MA) catalyst were prepared by the incipient wetness impregnation of the MA support with a solution of nitrates. Precursors included $Ni(NO_3)_2.6H_2O$ (Merck 99%), $Ce(NO_3)_3.6H_2O$ (Honeywell Fluka$^{TM}$, Charlotte, NC, USA 99%), and $Co(NO_3)_3.6H_2O$ (Sigma-Aldish 99%). Then, the wet solid cake was dried at 60 °C for 24 h and calcined at 650 °C for 5 h.

### 3.2. Catalyst Characterization

The textural properties were characterized by $N_2$ adsorption-desorption isotherms at −196 °C employing a BEL-Japan BELSORP-mini II (BEL JAPAN, INC., Osaka, Japan). Samples were previously in the nitrogen flow at 350 °C for 4 h. The specific surface area and total pore volume were calculated by the Brunauer–Emmett–Teller (BET) method, whereas the pore size distribution was evaluated by the Barrett-Joyner-Halenda (BJH) method.

The crystalline phase compositions were determined by the X-ray powder diffraction (XRD) patterns using a Model D8 Discover (Bruker AXS, Billerica, MA, USA) with Cu-Kα radiation operating at 40 kV and 40 mA using a speed of 0.02°/min.

The metal particle size and metal distribution on the fresh catalysts were investigated by transmission electron microscope (TEM) on a JEOL JEM-2010 (JEOL Ltd., Welwyn Garden City, England) operating at an acceleration voltage of 200 keV. The catalyst powders were suspended in absolute ethanol, sonicated, and deposited on a carbon copper grid (300 Mesh). Grids were dried before TEM characterization.

The reducibility and metal-support interaction were analyzed by the $H_2$ temperature programmed reduction ($H_2$-TPR) on a BELCAT-basic system (BEL JAPAN, INC., Osaka, Japan) equipped with a thermal conductivity detector (TCD). A 50-mg sample was pretreated in an Ar flow at 220 °C for 40 min and cooled to room temperature. Then, 5% $H_2$/Ar gas mixture was passed through the catalyst from room temperature to 900 °C at the rate of 10 °C/min.

The basic site distribution was evaluated using the $CO_2$ temperature programmed desorption ($CO_2$-TPD) data obtained on the BELCAT-basic system. The calcined catalyst (50 mg) was pre-reduced in an $H_2$ pure at 620 °C for 2 h and, subsequently, cooled to room temperature in a He flow. The isothermal adsorption of $CO_2$ gas was performed for 30 min before the sample was flushed in a He flow. The $CO_2$ desorption was then monitored by TCD in a He flow from room temperature to 800 °C with a ramping rate of 10 °C/min.

The carbon deposition on the spent catalysts was analyzed using thermogravimetric analysis (TGA) on a Model TGA/DSC1 (METTLER TOLEDO, Columbus, OH, USA). A quantity of 15 mg of the spent catalysts was combusted under air flow of 30 mL/min from room temperature to 800 °C at a heating rate of 10 °C/min.

### 3.3. Catalytic Activity Test

CSCRM tests were performed in a fixed-bed reactor at 620 °C under atmospheric pressure for 6 h. A 200-mg catalyst was in situ reduced at 620 °C in a pure $H_2$ flow of 30 mL/min for 6 h. The volumetric ratio of $CH_4$:$CO_2$:$H_2O$:$N_2$ = 3:5:2.26:4 (ultra-low S/C of 0.28) was fed with the flow rate of 60 mL/min. It should be noted that $N_2$ was used as a carrier gas of the steam. After passing through a glass cold trap, the composition of outlet gases was examined using an on-line gas chromatograph (Agilent

GC7890A Agilent, Santa Clara, CA, USA) equipped with TCD. The reactant conversions and the $H_2$/CO ratio were calculated as:

$$\% \ CH_4 \ conversion = \frac{\text{Flow rate } CH_{4,in} - \text{Flow rate } CH_{4,out}}{\text{Flow rate } CH_{4,in}} \times 100 \qquad (11)$$

$$\% \ CO_2 \ conversion = \frac{\text{Flow rate } CO_{2,in} - \text{Flow rate } CO_{2,out}}{\text{Flow rate } CO_{2,in}} \times 100 \qquad (12)$$

$$\frac{H_2}{CO} \ ratio = \frac{\text{Flow rate of } H_{2,out}}{\text{Flow rate of } CO_{,out}} \qquad (13)$$

## 4. Conclusions

The 5 wt% Ni–5 wt% Ce/$Al_2O_3$-MgO (5Ni5Ce/MA), 5 wt% Ni–5 wt% Co/$Al_2O_3$-MgO (5Ni5Ce/MA), and 10 wt% Ni/$Al_2O_3$-MgO (10Ni/MA) catalysts have been synthesized, characterized by a number of analytical techniques, and tested for their catalytic performance in combined steam and $CO_2$ reformation of methane (CSCRM) with the low S/C ratio. The characterization results revealed that the appearance of $CeO_2$ and Co in the Ni/MA catalyst improved the metal dispersion, resulting in the smaller Ni nanoparticle size. The $CeO_2$ promoter raises the number of all types of basic sites, whereas the Co promoter slightly increases the number of medium basic sites because of its oxophilic property.

Under the low S/C ratio condition of CSCRM process at 620 °C, although the highest activity toward $CH_4$ conversion was obtained from the 10Ni/MA catalyst, the 10Ni/MA catalyst exhibited the highest carbon accumulation. The O* species at the surface were insufficient to remove carbon deposits. The addition of each promoter ($CeO_2$ and Co) enhanced the O* species at the surface by improving the activation-dissociation of the different co-oxidant ($H_2O$ and $CO_2$), suppressing the carbon accumulation and resulting in the opposite trend of the $H_2$/CO ratio. $CeO_2$ promoted the $H_2O$ activation-dissociation, increasing the $H_2$/CO ratio. Co promoted the $CO_2$ activation-dissociation, decreasing the $H_2$/CO ratio.

**Author Contributions:** Conceptualization, N.D., T.R., S.T., and M.P. Data curation, N.D. and M.P. Investigation, N.D., T.R., S.T., and M.P. Methodology, N.D. and M.P. Formal analysis, N.D. and M.P. Funding support, T.S. Writing—original draft, N.D. and M.P. Writing—review and editing, N.D., T.S., D.S.A.S., and M.P. Supervision, D.S.A.S., S.T., and M.P. All authors have read and agreed to the published version of the manuscript.

**Funding:** This work was supported by the Thailand Science Research and Innovation (TSRI) via Research and Researchers for Industries (RRI) with the PTT Public Company Limited (grant number PHD59I0027).

**Conflicts of Interest:** The authors declare no conflict of interest.

## Nomenclature

| | |
|---|---|
| BET | Brunauer–Emmett–Teller |
| BJH | Barrett–Joyner–Halenda |
| CSCRM | Combined steam and $CO_2$ reforming of methane |
| $CO_2$-TPD | $CO_2$ temperature programmed desorption |
| $H_2$-TPR | $H_2$ temperature programmed reduction |
| IUPAC | International Union of Pure and Applied Chemistry |
| RWGS | Reverse water gas shift |
| S/C ratio | Steam-to-carbon ($H_2O$/($CH_4$ + $CO_2$)) ratio |
| TCD | Thermal conductivity detector |
| TEM | Transmission electron microscope |
| TGA | Thermogravimetric analysis |
| XRD | X-ray diffraction |

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
