# Peer review of "The Effects of CeO2 and Co Doping on the Properties and the Performance of the Ni/Al2O3-MgO Catalyst for the Combined Steam and CO2 Reforming of Methane Using Ultra-Low Steam to Carbon Ratio"

_catalysts, doi:10.3390/catal10121450_

Round 1

Reviewer 1 Report

The manuscript catalysts-1024099 report interesting findings on CeO2 / Co doped Ni/Al2O3-MgO catalyst for combined steam and CO2 reforming of methane. The scientific finding of the manuscript is ok, but the language of the manuscript needs improvement (i.e. check highlighted parts in the attached file)

Scientific comment:

1-the labeling of the peaks in H2-TPR is mainly based on the references. It should be considered that reduction temperature could be affected by changing the composition of the materials (here new complex material is prepared) and therefore I suggest change the text about peak assignment and make the sentences milder (i.e. you can mention peak at 400 oC can tentatively assigned to …, specially about Fig. 4 b).

Author Response

Response to Reviewer 1 Comments

Comments and Suggestions for Authors

Point 1: The manuscript catalysts-1024099 report interesting findings on CeO2 / Co doped Ni/Al2O3-MgO catalyst for combined steam and CO2 reforming of methane. The scientific finding of the manuscript is ok, but the language of the manuscript needs improvement (i.e. check highlighted parts in the attached file)

Response 1: We thank the reviewer very much for pointing this out. We have checked highlighted parts in the attached file from reviewer 1 and revised through the manuscript and accordingly.

Line 24: change from S/C ratio to steam-to-carbon ratio (S/C ratio)

Line 41-43: change from “Although CSCRM can control an H2/CO ratio of 2 in the syngas product by the inlet feed composition [2,5,6]. The obstacles for a commercial CSCRM consist of the catalyst deactivation and the energy consumption.” to “Although CSCRM can control an H2/CO ratio of 2 in the syngas product by the inlet feed composition [2,5,6], the obstacles for a commercial CSCRM consist of the catalyst deactivation and the energy consumption.”

Line 53-54: change from “it was founded that cerium (Ce) in the form of ceria (CeO2) is partially reduced (Ce4+ to Ce3+) and creates oxygen vacancy sites.” to “Among several metals, the oxygen vacancy sites can be created when Ce4+ in the oxide form of cerium (CeO2) transforms to Ce3+.”

Line 57: change from suggested that “Ni2+ is incorporated into the lattice of CeO2 to form the Ce1−XNiXO2 (0.05 ≤ x ≤ 0.5) solid oxide solution” to “Ni2+ ions can be incorporated into the lattice of CeO2

 Line 70: change from “influences to” to “deducts

Line 91-92: change from “The CSCRM with ultra-low S/C were demonstrated at 620oC for 6 h under the atmospheric pressure.” to “The CSCRM performances of all catalyst samples were investigated at 620°C under atmospheric pressure.” 

Line 103: change from “obtained from” to “observed on

Line 105: change from “was” to “were

Line119: change from “to” to “with

Line121: change from “resist” to “reduces

Line174-175: change from “the somewhat differencet conversions for CH4 and CO2 and  low H2/CO ratio (<1).” to “the different conversions for CH4 and CO2 with low H2/CO ratio (<1).”

Line176: change from “implied” to “referred

Line178: not change “indicated

Line180: change from “mentions” to “reflected

Line182-183: change from “The 5Ni5Ce/MA catalyst exhibited the lowest CO2 conversion with the highest H2/CO ratio was increased about 1.6 in CSCRM reaction” to “The 5Ni5Ce/MA catalyst showed the lowest CO2 conversion with the highest H2/CO ratio

Line 305: change from “influenced” to “raises the number of

Line 311-315: change from “The behavior of each promoter (CeO2 and Co) attracted the different co-oxidant (H2O and CO2) absorption which affected the H2/CO ratio. The CeO2 promoter encouraged the H2O association-dissociation, resulting in the high H2/CO ratio. As the Co promoter boosted the CO2 association-dissociation, the existence of Co promoter involved the low H2/CO ratio under the CSCRM process with the low S/C ratio. Even though bimetallic catalysts showed the lower CH4 conversion, the addition of CeO2 and Co promoters on the Ni/MA catalyst enhance the H2O and CO2 association-dissociation, respectively, via the medium basic sites, evidenced by the suppression of carbon formation.” to “The addition of each promoter (CeO2 and Co) enhanced the O* species at the surface by improving the activation -dissociation of the different co-oxidant (H2O and CO2), suppressing the carbon accumulation and resulting in the opposite trend of the H2/CO ratio. CeO2 promoted the H2O activation-dissociation, increasing H2/CO ratio. Co promoted the CO2 activation-dissociation, decreasing H2/CO ratio.”

Scientific comment:

Point 2: the labeling of the peaks in H2-TPR is mainly based on the references. It should be considered that reduction temperature could be affected by changing the composition of the materials (here new complex material is prepared) and therefore I suggest change the text about peak assignment and make the sentences milder (i.e. you can mention peak at 400oC can tentatively assigned to …, specially about Fig. 4 b).

Response 2: We agree with the reviewer. In the revised manuscript, we changed the text about peak assignment and made the sentences milder specially about Fig. 4b as seen in the catalyst preparation part (line 137-141). “The lowest temperature peak is preliminarily ascribed to the oxygen mobility on the surface catalyst that takes place the substitution of Ni ion incorporated with CeO2 at the surface and the reduction of Ce4+ located on the surface [24,36,37]. The second and third peaks can be tentatively assigned to the reduction of Ni-CeOx solid solution.”

Reviewer 2 Report

The Manuscript deals with the preparation, the characterization of Ni-based catalysts doped with CeO2 and Co and the study of activity in combined combined steam and CO2 reforming of methane.

The topic is surely current and interesting. Many experimental data are reported, and the adopted methodologies are appropriate. The presented work has high clarity in the objectives and performed analysis. The topic is in good agreement with the journal.

The manuscript therefore can deserve publication. Nevertheless, some few observations should be addressed and amended by the Authors prior to final acceptance.

Comments:

1) The authors, starting from the title, propose to discuss the role of CeO2 and Co in the catalysts prepared in the reaction of steam and CO2 reforming. However, this fact is not sufficiently emphasized in the text. The authors should accompany the reader in understanding the effect of these two metals in the reaction mechanism, justifying the results obtained in the light of the results of the characterizations. 

2) Regarding with the synthesis of the catalysts, the sol-gel method was used for the supports and impregnation for the final catalysts. However, the description of the procedures followed to obtain supports and catalysts is somewhat incomplete, the authors should enrich this part with more details.

3) The authors state that catalysts with CeO2 and Co are characterized by a higher specific area and better dispersion, compared to the catalyst without. But between these three systems there is a difference in the Ni content. For a complete picture the authors should also indicate the dopant-free solid with the same Ni content.

4) An important aspect in this reaction concerns the deactivation by the Coke. The authors report TGA values ​​to discuss carbon deposits, but it is unclear whether the proposed catalysts are stable or not.

5) The relative errors related to conversion values and characterization measures of catalysts are missing. These data are important to understand the scattering of some results reported in the tables and figures.

Author Response

Response to Reviewer 2 Comments

Point 1: The authors, starting from the title, propose to discuss the role of CeO2 and Co in the catalysts prepared in the reaction of steam and CO2 reforming. However, this fact is not sufficiently emphasized in the text. The authors should accompany the reader in understanding the effect of these two metals in the reaction mechanism, justifying the results obtained in the light of the results of the characterizations.

Response 1: We thank very much for your suggestions that truly improve our article. We have added the simplified reaction mechanism and also discussed how the mechanism relates with the results of the characterizations. The additional discussion in the revised manuscript at the results and discussion section after the TGA part (after line 212) would accompany the reader in understanding the effect of these two metals in the reaction mechanism.

Revise

line 219-237: A simplified reaction mechanism of CSCRM to produce synthesis gas on a Ni-based catalyst consists of three element reactions: CH4 dissociation/decomposition (Eq. 7), H2O activation-dissociation (Eq. 8) and CO2 activation-dissociation (Eq. 9). CH4 dissociation/decomposition produces carbon surface (C*) and H2. CO2 and H2O activation-dissociation generate oxygen surface (O*) with CO and H2, respectively. The C* on the surface can be then removed by reacting with the O* surface to form CO (Eq. 10). Therefore, the catalytic behavior of the catalyst for the reaction steps can be tentatively evaluated by conversions of reactants, H2/CO ratio, and the quantity of coke formation. The activity toward the CH4 dissociation/decomposition step directly relates to CH4 conversion. The H2O and CO2 activation-dissociation can be preliminary predicted using H2/CO ratio with the amount of coke. If the high H2/CO ratio with low carbon accumulation are occurred, the catalyst is selective for H2O activation-dissociation. Likewise, the catalyst is selective for CO2 activation-dissociation when the low H2/CO ratio and low carbon accumulation are founded. According to the results under the low S/C ratio condition of CSCRM process, it suggested that 10Ni/MA is the most active catalyst for CH4 dissociation/decomposition. CeO2 in 5Ni5Ce/MA and Co in 5Ni5Co/MA boots the H2O and CO2 activation-dissociation, respectively.

CH4 + * ® 2H2 + C*                                                                           (7)

CO2 + * ® CO + O*                                                                           (8)

H2O + * ® H2 + O*                                                                           (9)

C*  + O* ® CO                                                                                (10)

Point 2: Regarding with the synthesis of the catalysts, the sol-gel method was used for the supports and impregnation for the final catalysts. However, the description of the procedures followed to obtain supports and catalysts is somewhat incomplete, the authors should enrich this part with more details.

Response 2: We thank for your comment. We have added more details for a better procedure of the catalyst synthesis to provide as seen in the catalyst preparation part (line 214-220).

Revise

line 240-248: The Al2O3-MgO (MA) was synthesized by the sol-gel method. The mixture of aluminium isopropoxide (Acros OrganicsTM 98%), nitric acid (CARLO ERBA Reagents 65%), magnesium ethoxide (Sigma-Aldrich 98%) and distilled water were refluxed at 80°C for 20 h. The resulting gel was dried at 60oC for 24 h and calcined at 800oC for 6 h. The MA support was crushed and sieved to 355-710 mm. The 10 wt% Ni/Al2O3-MgO (10Ni/MA), 5 wt% Ni-5 wt% Ce/Al2O3-MgO (5Ni5Ce/MA) and 5 wt% Ni-5 wt% Co/Al2O3-MgO (5Ni5Co/MA) catalyst were prepared by the incipient wetness impregnation of the MA support with a solution of nitrates. Precursors included Ni(NO3)2.6H2O (Merck 99%), Ce(NO3)3.6H2O (Honeywell FlukaTM 99%) and Co(NO3)3.6H2O (Sigma-Aldish 99%). Then, the wet solid cake was dried at 60oC for 24 h and calcined at 650oC for 5 h

Point 3: The authors state that catalysts with CeO2 and Co are characterized by a higher specific area and better dispersion, compared to the catalyst without. But between these three systems there is a difference in the Ni content. For a complete picture the authors should also indicate the dopant-free solid with the same Ni content.

Response 3: We thank you for this point of suggestion. In this research article, we compare all catalysts in the case of similarly total metal content (10Ni/MA, 5Ni5Ce/MA, and 5Ni5Co/MA). It would have been interesting to compare the dopant-free solid with the same Ni content. We understand your concerning. However, we apologize indeed that we cannot complete experiments of the 5Ni/MA catalyst within four days according to the time for catalyst preparation and the different experiments.

Point 4: An important aspect in this reaction concerns the deactivation by the Coke. The authors report TGA values to discuss carbon deposits, but it is unclear whether the proposed catalysts are stable or not.

Response 4: This suggestion helps to improve our article. We also understand this reaction concerning. Therefore, we rewrote the TGA part to relate the results with the deactivation by the coke. This revision should clarify that the CeO2 and Co promoters enhance the carbon tolerance of Ni/MA catalyst (line 198-209)

Revise

line 203-216: The quantity and the types of carbon deposition on the spent catalysts were elucidated using the TGA results (Figure 7). The percentage of weight loss directly relates to the amount of carbon deposition. TGA profiles of the spent catalysts represented the physically adsorbed water and amorphous carbon (≤ 250oC), the graphitic carbon (250oC - 450oC) and the carbon filament(≥ 450oC) on the surface [18,52–54]. The last two types, which are not easily oxidizable, have been considered as the major reason for the catalyst deactivation. The total weight loss of spent 10Ni/MA, 5Ni5Ce/MA and 5Ni5Co/MA catalysts were 17%, 11% and 12%, respectively. Although the main type of coke in all catalysts was the graphitic carbon, the highest temperature of carbon removal was found from the spent 10Ni/MA catalyst. It implied that carbon deposition was formed more easily via the CH4 dissociation/decomposition on the Ni/MA catalyst. The coke deposited can be deduced by the oxygen intermediate in H2O activation-dissociation for 5Ni5Ce/MA and oxygen intermediates in CO2 activation-dissociation for 5Ni5Co/MA. Consequently, with Ce and Co promoters, the carbon tolerance of Ni/MA catalyst was improved, requiring only a small amount of steam added in the feed of the methane reforming in order to prevent carbon deposition [55-57]

Point 5: The relative errors related to conversion values and characterization measures of catalysts are missing. These data are important to understand the scattering of some results reported in the tables and figures.

Response 5: We thank for your comment. The systematic error of N2 adsorption-desorption is about ± 5%. As the TPD and TPR techniques are tested in the same instrument, the systematic error is about ± 1% for temperature and about ± 8 % for the quantity of gaseous substances. For the relative errors of conversion values is about ± 3%. We have added the systematic error under Table 1 and Figure 6, respectively.

Revise (line127-129)

Table 1. The properties of 10Ni/MA, 5Ni5Ce/MA and 5Ni5Co/MA catalysts.

N2 adsorption-desorptiona

H2-TPRb

Deconvolution of the CO2-TPDb

(mmol/gcat)

Catalyst

Surface area

)m2/g(

Total pore volume

)cm3/g(

H2 uptake

(mmol/gcat)

Weak

Medium

Strong

Total basicity

10Ni/MA

97

0.20

11.9

0.04

0.02

0.13

0.19

5Ni5Ce/MA

111

0.24

4.3

0.02

0.03

0.17

0.22

5Ni5Co/MA

108

0.22

10.8

0.03

0.03

0.10

0.16

a The systematic error of N2 adsorption-desorption is  ± 5%

b The systematic error is  ± 1% for temperature and  ± 8 % for the quantity of gaseous substances.

(line 201-202)

Figure 6. The CH4 (a) and CO2 conversions (b) for CSCRM and H2/CO ratios (c) of 10Ni/MA, 5Ni5Ce(T)/MA, 5Ni5Ce/MA catalysts at 620oC under ambient pressure (the relative error is 3%)
